# Physical-Chemical Characterization of Different Carbon-Based Sorbents for Environmental Applications

**DOI:** 10.3390/ma15207162

**Published:** 2022-10-14

**Authors:** Simone Marzeddu, María Alejandra Décima, Luca Camilli, Maria Paola Bracciale, Virgilio Genova, Laura Paglia, Francesco Marra, Martina Damizia, Marco Stoller, Agostina Chiavola, Maria Rosaria Boni

**Affiliations:** 1Department of Civil, Constructional and Environmental Engineering (DICEA), Faculty of Civil and Industrial Engineering, Sapienza University of Rome, Via Eudossiana 18, 00184 Rome, Italy; 2Department of Chemical Engineering Materials Environment (DICMA), Faculty of Civil and Industrial Engineering, Sapienza University of Rome, Via Eudossiana 18, 00184 Rome, Italy

**Keywords:** activated carbon, adsorbent, biochar, carbon-based materials, characterization methods, circular economy, environmental processes, wastewater treatment

## Abstract

Biochar has been used in various applications, e.g., as a soil conditioner and in remediation of contaminated water, wastewater, and gaseous emissions. In the latter application, biochar was shown to be a suitable alternative to activated carbon, providing high treatment efficiency. Since biochar is a by-product of waste pyrolysis, its use allows for compliance with circular economics. Thus, this research aims to obtain a detailed characterization of three carbonaceous materials: an activated carbon (CARBOSORB NC 1240^®^) and two biochars (RE-CHAR^®^ and AMBIOTON^®^). In particular, the objective of this work is to compare the properties of three carbonaceous materials to evaluate whether the application of the two biochars is the same as that of activated carbon. The characterization included, among others, particle size distribution, elemental analysis, pH, scanning electron microscope, pore volume, specific surface area, and ionic exchange capacity. The results showed that CARBOSORB NC 1240^®^ presented a higher specific surface (1126.64 m^2^/g) than AMBIOTON^®^ (256.23 m^2^/g) and RE-CHAR^®^ (280.25 m^2^/g). Both biochar and activated carbon belong to the category of mesoporous media, showing a pore size between 2 and 50 nm (20–500 Å). Moreover, the chemical composition analysis shows similar C, H, and N composition in the three carbonaceous materials while a higher O composition in RE-CHAR^®^ (9.9%) than in CARBOSORB NC 1240 ^®^ (2.67%) and AMBIOTON^®^ (1.10%). Differences in physical and chemical properties are determined by the feedstock and pyrolysis or gasification temperature. The results obtained allowed to compare the selected materials among each other and with other carbonaceous adsorbents.

## 1. Introduction

The new European circular economic action plan aims to speed up the transition to a circular economy and to decarbonize hard-to-abate sectors [1,2,3]. In the circular economy, resources remain in a circle, minimizing waste, and making it possible to recycle them in subsequent industrial applications [4].

The need of a transition from a linear to a circular economy fostered the study of potential reuse of scraps and wastes, eventually modified to enhance their properties [2,5], i.e., agricultural wastes can be used to produce by-products such as fertilizers, energy, and new materials [6].

In environmental processes, these new materials with high efficiency and economic-environmental sustainability have been extensively researched [7,8,9,10,11,12,13], whereas limiting these scraps and wastes and transforming them into products that implement the principles of reuse, repair, and recycling can help local economies to generate profits and reduce environmental damage [14]. Through the increasingly in-depth knowledge of materials, it is possible to avoid production processes that are harmful to the environment and produce further waste [15]. Furthermore, the possible reuse of all by-products of the processes [16,17] can be evaluated through the lifecycle assessment [18,19,20,21].

Particularly, solid by-products deriving from the pyrolysis of organic substances received the attention of the research community due to their peculiar characteristics, i.e., high porosity, electrical surface charge, and wide active surface area, making them favorable for the sorption of various contaminants from water [22,23], wastewater, and contaminated soils [24,25,26,27,28], such as activated carbon and biochar [29].

Biochar (BC), mainly used as a soil improver, is a carbonaceous solid by-product deriving from pyrolysis, having a high porosity [30,31,32]. Biochar’s characteristics are largely determined by the properties of the original feedstock and the carbonization conditions [33,34,35].

Pyrolysis produces the thermochemical conversion of biomass [36,37] and involves a series of processes such as condensation, decarboxylation, demethylation, oxidation, and reduction [38]. The pyrolysis process takes place at high temperatures (>300 °C) under the exclusion of oxygen. It generates three types of final products: a gaseous flux, referred to as syngas, used to generate energy or for the production of fine chemicals [39]; a liquid waste, referred to as bio-oil [40], used to produce electricity and heat in small to medium stationary applications; a solid product, referred to as biochar.

Depending on the temperature, three different types of pyrolysis can be distinguished:Slow pyrolysis, which is characterized by low heating rates, relatively low temperatures (300–400 °C), and long residence times [41]. This pyrolysis technique is mostly used to produce high quantities of biochar (yields of about 30%).Fast pyrolysis, which is characterized by high thermal gradients (≈100 °C/s), high temperatures (400–700 °C), and very short residence times (seconds) [42]. This pyrolysis technique is used to obtain high bio-oil yields (over 50%).Conventional pyrolysis, characterized by moderate heating rates (≈20 °C/s) and equally moderate reaction temperatures (less than 600 °C), with residence times ranging from 10 s to 10 min [43]. This type of pyrolysis gives rise to comparable quantities of char, gas, and liquid.

Biochar’s main use is as a soil amendment [44,45,46]; however, it has also been used as an alternative adsorbent to activated carbon since it better complies with the aims of the circular economy [47,48,49].

Activated carbons (AC) have been used at a much larger scale than BC in several process applications, including treatment of drinking water, deodorization and aroma removal, industrial flue gas cleaning, and air conditioning.

They have particularly favorable characteristics for adsorption processes, due to their properties such as high porosity, large surface area, high surface reactivity, and ease of compaction into a packed bed [50,51,52,53,54]. It can be classified into powdered activated carbon (PAC) and granulated activated carbon (GAC), among others [55].

AC is produced from different sources (coal, peat, wood, or lignite), through two methods, physical activation and chemical activation [56], each one carried out at different temperatures [28,57].

Physical activation consists of two phases: carbonization and oxidation. In particular, after dehydration to eliminate water, the material is subjected to a temperature of around 500 °C in the absence of air [58,59]. The addition of metal chlorides in the first phase of the process favors the development of pores [60]. The subsequent oxidation phase is usually carried out using steam [61], although air (less frequently CO_2_) is sometimes chosen, at temperatures ranging between 600 and 1200 °C [62,63]. During the oxidation phase, gases erode the surface of the coal, developing a vast internal network of pores [64].

On the other hand, chemical activation by chemical reagents consists in a unique phase. The precursors, first to be carbonized, are imprinted with a dehydrating chemical agent such as ZnCl_2_, H_3_PO_4_, NaOH, KOH, ZnCl_2_, K_2_S, KCNS, or KMnO_4_ [65,66]. After impregnation, carbonization is conducted at different temperatures depending on the chosen activating agent, but generally at a lower temperature than for physical activation, ranging from 300 to 800 °C.

Through the increasingly in-depth knowledge of materials, it is possible to avoid production processes that are harmful to the environment and produce further waste [15]. Furthermore, the possible reuse of all by-products of the processes [16,17] can be evaluated through the lifecycle assessment [18,19,20,21].

To determine the possible uses of these new materials under safe and sustainable conditions, an accurate chemical characterization is important to determine the composition and structural properties which are strictly related to the adsorption capacity [67,68,69,70]. Furthermore, knowing the chemical composition of a material guarantees the required quality and safety of the final product [71], with attention to the environment and health of consumers [72,73,74].

Particularly, the characterization of biochar and activated carbon allows knowing the potential uses of biochar and comparing them with activated carbon’s characteristics. However, only a few articles have fully characterized activated carbon and biochar. Since biochar is a carbonaceous by-product with similar characteristics to activated carbon and since it is used mainly as a soil improver, it could also be used as an adsorbent in water treatment.

Thus, the present work reports a full physical-chemical characterization of three different carbon-based materials: one activated carbon (CARBOSORB NC 1240^®^) and two commercial biochars (RE-CHAR^®^ and AMBIOTON^®^). To the authors’ knowledge, these three carbonaceous materials have not yet been fully characterized together.

The novelty is to compare their characteristics to evaluate the possibility of replacing activated carbon with biochar for the environmental applications; specifically, since BC is a carbonaceous by-product with characteristics similar to AC, it could possibly also be used as an adsorbent in water treatment.

## 2. Materials and Methods

### 2.1. Activated Carbon and Biochar

The following three carbonaceous materials were characterized: a commercial activated carbon named “CARBOSORB NC 1240^®^”, and two Italian biochars named “AMBIOTON^®^” and “RE-CHAR^®^”, respectively.

CARBOSORB NC 1240^®^, supplied by the COMELT company, is a granular activated carbon, produced from coconut shells by physical activation (gasification temperature >900 °C). The CARBOSORB NC 1240^®^ complies with the UNI EN 12915-1: 2004 standard “Granular Activated Carbon for use for the treatment of water for human consumption”.

AMBIOTON^®^, supplied by the Laterizi Reato S.R.L. company, is a biochar obtained through a pyrolysis process at T = 700–750 °C from large pieces of semi-de-husked wood [75].

RE-CHAR^®^, produced by Record Immobiliare SRL, is a biochar derived from virgin coniferous wood (mainly pine), through a pyrolysis or gasification process (T = 600–750 °C), and classified as Biochar class A1 according to ISO 17225-4: 2014 from forestry [76,77]. In Boni et al. and Chiavola et al.’s research, the technical data of both biochars are described in detail [78,79].

### 2.2. Chemicals and Reagents

All chemicals and reagents used during laboratory tests were of analytical grade.

### 2.3. Physical Characterization

Methods and processes applied in the physical characterization of the materials are described in the following sections.

#### 2.3.1. Particle Size Distribution

Particle size distribution (PSD) was determined by sieving. For each material, about 30 g was sieved by different diameter size sieves (850, 600, 500, 355, 300, 212, 180, 90, 75, 53, 45, and 38 µm), and the fraction of solids retained by each was weighed and then used to calculate the percentage of material withheld from the total. The sieving was carried out in the most appropriate range of sizes considering technical datasheets of the carbonaceous materials.

#### 2.3.2. Density, Bulk Density, and Specific Weight

Bulk density (ρ_b_, kg/m^3^), density (ρ, kg/m^3^), and specific weight (γ, N/m^3^) have been estimated according to ASTM Standards [80] and CEFIC [81].

#### 2.3.3. Specific Surface Area, Pore Volume, and Average Pore Size

The specific surface area (S_BET_, m^2^/g), pore volume (V_T_, cm^3^/g), and average pore size (d, Å) were determined by nitrogen adsorption–desorption isotherms acquired at −196 °C using a 3Flex analyzer (Micromeritics Instrument Corporation).

The adsorption–desorption isotherms were acquired in a relative pressure (*p*/*p*_0_) range from 0.01 to 0.99 [82]. Isotherm analyses were performed using the 3Flex Version 4.05 software. Samples were previously outgassed at 200 °C for 12 h.

The Brunauer–Emmett–Teller (BET) and Barrett–Joyner–Halenda (BJH) equations were used to determine the specific surface area, pore volume, and average pore size, respectively.

Initially, the BET Equation (1) was plotted to determine the monolayer absorbed gas volume (*v_m_*, cm^3^) and the BET isotherm constant (*C*, -), from the slope (Equation (2)) and intercept (Equation (3)) of the linear form (y = slope x + intercept):(1)1v [(p0p)−1]=(C−1)vm C(pp0)+1vm C
(2)$$slope=\frac{\left(C-1\right)}{v_{m}\;C}$$
(3)$$intercept=\frac{1}{v_{m}\;C}$$
where *v* is the volume filled by multilayer adsorption on the external surface (cm^3^) [83].

From *v_m_*, we can determine the specific surface area (m^2^/g), through Equation (4):(4)$$S_{BET}=\frac{v_{m}\;s\;N}{m_{1}\;V}$$
where *s*, *N*, *m*_1_, and *V* are the cross-sectional area of the adsorbed gas molecule (m^2^), the molar volume of adsorbed gas, the mass of the sample (g), and the Avogadro’s number, respectively.

The method of BJH is a procedure for calculating pore size distributions from experimental isotherms using the Kelvin model of pore filling, and it applies only to the mesopore and small macropore size ranges [84].

It assumes that the hole type is a cylindrical hole, which is only applicable to the specified pore size range and underestimates the pore size, which can easily lead to significant deviations (when the pore size is <5 nm, the deviation can be up to 25%), mainly suitable for mesoporous columnar models larger than 5 nm [85].

The Kelvin equation is based on thermodynamics and has the following expression of Equation (5) [86]:(5)$$\ln\left(\frac{p}{p_{0}}\right)=-\frac{2\;\gamma_{L}\;V_{L}}{r_{K}\;R\;T}\cos\theta$$
where, *p*/*p*_0_ is the relative pressure, *γ_L_* and *V_L_* are the surface tension and molar volume of the liquid, respectively, *θ* is the contact angle of the liquid in the micropore, *R* and *T* are the universal gas constant and absolute temperature, respectively, and *r_K_* is the Kelvin radius (or critical radius) [87]. The Dollimore and Heal method calculates the true radius of the micropore (*r*) as the summation of *r_K_* and the adsorbed layer thickness on the pore wall (*t*), when condensation occurs at a given relative pressure, through the following Equation (6) [88]:(6)$$r=r_{K}+t$$

The Harkin-Jura and Halsey equations were used to predict *t*, as shown below with Equations (7) and (8) [89]:(7)t=110[13.99 ln100.34 ln10−ln(pp0)]12

In the Equation (7), the adsorbed layer thickness is well-matched with the measured value when *p*/*p*_0_ is smaller than 0.3; however, it becomes unsatisfying when the relative pressure is greater than 0.5 [89].
(8)t=τ[52.303 log(p0p)]13
where τ is the thickness of the single adsorption layer. When the adsorbate is N_2_, τ = 0.354 nm, and the equilibrium relationship between the thickness of the adsorption layer and the pressure can be obtained by Equation (9) [86]:(9)t=0.354[−5ln(pp0)]13

The ISO 15901 [90] provides clear constraints on the use of BJH: pores are rigid and have regular shapes (i.e., cylindrical), no micropores exist, and the pore size distribution is discontinuous beyond the maximum porosity that can be measured by this method, that is at the highest relative pressure, where all measured pores are filled.

#### 2.3.4. Moisture Content, Volatile Matter, Ash Content, and Fixed Carbon

Moisture (M, %), ash content (A, %), volatile matter (VM, %), and fixed carbon (FC, %) were determined by proximate analysis of biochar samples using thermogravimetric analysis (TGA), as reported elsewhere [91].

All experiments were performed on a TA Instruments SDTQ600 thermobalance (TA Instrument). To avoid limitations on mass and heat transfer, about 10 mg of sample and uncovered platinum crucibles were used. TG curves were obtained under nitrogen (N_2_) and air flow of 45 mL min^−1^, increasing the temperature from the room value up to 600 °C and with a heating rate of 10 °C min^−1^.

All analyses were initially conducted under an inert N_2_ flow to prevent oxidation and to determine the moisture and volatile material concentration. Then, dry air was used to determine the ash content.

The moisture content (*M*) was determined by heating the sample at T ≈ 105 °C in a N_2_ atmosphere until a constant weight was reached. The moisture content was obtained from the following Equation (10):(10)$$M\;\left(\%\right)=\frac{\left(m_{1}-m_{2}\right)}{m_{1}}\;\cdot \;100$$
where *M* is the difference between the initial mass (*m*_1_) of the sample and the constant mass (*m*_2_) at T ≈ 105 °C.

Volatile matter (*VM*) was determined as weight loss due to heating from T ≈ 105 °C to T ≈ 600 °C in an atmosphere of N_2_, according to the following Equation (11):(11)$$VM\;\left(\%\right)=\frac{\left(m_{2}-m_{3}\right)}{m_{1}}\;\cdot \;100$$
where *m*_3_ (mg) is the mass of the sample, when the heating temperature reaches T ≈ 600 °C.

Ash (*A*) is the residual inorganic matter determined after combustion at T ≈ 600 °C under dry air, and it was obtained from Equation (12):(12)$$A\;\left(\%\right)=\frac{m_{4}}{m_{1}}\;\cdot \;100$$
where *m*_4_ (mg) is the residual mass at the end of the analysis. Subsequently, the amount of fixed carbon (*FC*) was determined by Equation (13):(13)FC (%)=100−M−VM−A

#### 2.3.5. Conductivity

The conductivity (G, Ω^−1^) was determined using a conductivity meter (FiveEasy FE30, Mettler Toledo™).

#### 2.3.6. Raman Spectroscopy

Raman spectroscopy is a technique based on the phenomenon of diffusion of electromagnetic radiation of an analyzed sample; in fact, the Raman spectral response is sensitive to the microscopic structure of the carbonaceous material [92].

For the acquisition of the Raman spectra, a micro-Raman dispersive spectrometer (SENTERRA, Bruker Optics) equipped with a laser CW diode-pumped solid-state laser (λ_0_ = 532 nm) and a 20x objective (Olympus B41) was used.

#### 2.3.7. Scanning Electron Microscopy and Energy-Dispersive X-ray Spectroscopy

The morphological studies were carried out by scanning electron microscopy (SEM, MIRA3 by Tescan™) equipped with an energy-dispersive X-ray analyzer (EDS) [93]. Before the analysis, the carbonaceous material was sputtered with a carbon coater (EM SCD005, Leica™).

#### 2.3.8. X-ray Diffraction

To study the crystalline structure of carbon-based materials, XRD (X-ray diffraction) analysis was performed using a Philips X’Pert diffractometer (PANalytical B.V.). The diffractometer works at 40 KV and 40 mA in a continuous scan mode in the 2θ range from 20° to 60°, with a step size of 0.02° and counting time of 2 s. The monochromatic radiation adopted was CuKα1.

### 2.4. Chemical Characterization

This section describes chemical methods used to characterize the carbonaceous materials.

#### 2.4.1. Elemental Analysis

Elemental analyses were carried out with a CHNOS elemental analyzer and an X-ray fluorescence (XRF) spectrometer (M4 Tornado, Bruker^®^).

The CHNOS elemental analyzer determines the percentage content of total hydrogen (H), nitrogen (N), carbon (C), and sulfur (S) using acetanilide as an internal standard [94], and oxygen content (O) was calculated by the difference. The quantities of C, H, N, and S were determined simultaneously by a gas chromatograph during the combustion of the samples [95,96], while the presence of O was assessed later by pyrolysis. To have a significant sample of the material to be analyzed, it was homogenized and finely ground, and the analyses were repeated three times and the average values were reported.

XRF is a technique that allows identifying the chemical elements that make up a sample, using X-rays and the radiation emitted following atomic excitation with appropriate energy [97]. The X radiation that hits the sample has a maximum energy of about 10 KeV. The information obtained comes from the most superficial layers of the sample, exactly the layers that the re-emitted characteristic radiation can cross.

A benchtop spectrometer (M4 Tornado, Bruker^®^), equipped with a Rh X-ray tube with poly-capillary optics as the X-ray convergence technique and an XFlash^®^ detector providing an energy resolution better than 145 eV and 5 filters, was utilized [10]. The whole spectra comprised 4096 channels with a spot size of approximately 30 μm. Spectrum energy calibration was performed daily before each analysis batch by using zirconium (Zr) metal (Bruker^®^ calibration standard).

#### 2.4.2. pH Analysis

For the determination of the pH value, a known quantity of carbonaceous material (i.e., 4.0 g) was mixed with a certain volume of boiled distilled water (i.e., 100 mL); subsequently, the temperature was increased up to the bubble point and kept constant for 5 min, after which, before it dropped below 60 °C, the pH of the supernatant was measured with a pH meter (Crison Instruments™) [81].

To determinate pH_PZC_, known quantities of carbonaceous material (0.01%, 0.1%, 1%, 5%, and 10% by mass) were added to 45 mL of ultrapure water at a known initial pH: as the quantity of carbonaceous material increases, the equilibrium pH of the solution tends progressively to an asymptotic value, which is, in fact, the point of zero charges (pH_PZC_) [98,99].

#### 2.4.3. Boehm Analysis

Functional groups present in carbonaceous material were identified by Boehm analysis [100,101].

For this, 1.0 g of material was put in contact with 50 mL of solutions at different concentrations (i.e., 0.05 M HCl, 0.05 M NaOH, 0.05 M Na_2_CO_3_, or 0.05 M NaHCO_3_) and shaken until constant pH (approximately 3 h) at room temperature, then filtered and titrated to evaluate the final concentration of the compounds, also measuring the pH value [102].

The indicators were bromothymol blue (C_27_H_28_Br_2_O_5_S) for NaHCO_3_, bromocresol green (C_21_H_14_Br_4_O_5_S) for Na_2_CO_3_, and phenolphthalein (C_20_H_14_O_4_) for NaOH [100,101].

The amounts of various types of acidic sites were calculated under the assumptions that NaOH neutralizes carboxylic, lactonic, and phenolic groups, Na_2_CO_3_ neutralizes carboxylic and lactonic groups, and NaHCO_3_ neutralizes only the carboxylic group. The number of surface basic sites was estimated from the amount of hydrochloric acid that reacted with the carbon [103].

#### 2.4.4. Cation and Anion Exchange Capacity

To determine the cation exchange capacity (CEC): 1.0 g of each carbonaceous material and 42 mL of BaCl_2_ were put in contact, stirred for 90 min, and subsequently subjected to an evaporation process in the oven for 24 h. After filtration with MgSO_4_, and stirring and filtration, 10 mL of each sample was recovered. The solution was prepared with 100 mL of H_2_O, 10 mL of NH_4_Cl (ammonium chloride), and eriochrome black (C_20_H_12_N_3_NaO_7_S) was used as an indicator.

This solution was titrated by adding ethylenediaminetetraacetic acid (EDTA) until the color change was noticed. A neutral sample was also analyzed, without the addition of carbon, which was composed of 100 mL of H_2_O, 10 mL of MgSO_4_, 10 mL of NH_4_Cl, and eriochrome black.

For the determination of the anion exchange capacity (AEC), 1.0 g of each carbon-based sorbent was adjusted to stable pH (within 2 weeks) using either NaOH or HBr. Once stability was achieved, 2.0 mL of KBr (1 M) was added and the solution was mixed for 48 h, then rinsed and filtered (0.45 μm) until conductivity was stabilized (i.e., 5 μS/cm).

Subsequently, to each carbonaceous material, transferred back into pyrex bottles, 2 mL of CaCl_2_ (2.5 M) and 50 mL of Milli-Q water were added, and the solutions were shaken for 48 h, diluted to 200 mL in a volumetric flask, and filtered with a syringe filter (0.45 μm). A known volume of each filtrate (i.e., 10 mL) was diluted to 100 mL, and finally the AEC was measured using bromide as an index anion [104].

#### 2.4.5. Methylene Blue and Iodine Index

The methylene blue index (MBI) is a quantitative measurement of the adsorption of methylene blue dye (MB) on the surface of carbon-based sorbents [105]. The MBI (g/kg) was measured by UV spectrophotometry.

Briefly, 0.1 g of carbonaceous material and 50 mL of a 10 mg/L solution of MB were put in contact for 24 h and stirred at 150 rpm. After that, the supernatant was collected after centrifugation (6000 rpm, 15 min) and the residual MB concentration was measured at λ_0_ = 668 nm, using a UV 1800 PC spectrophotometer (Agilent Technologies, Santa Clara, CA, USA).

The MBI was calculated as the difference between the initial MB concentration and the one measured in solution, divided by carbon-based sorbent mass concentration (mg of MB adsorbed per 1.0 g of carbonaceous material).

The iodine index (IV), or iodine adsorption, is the mass of iodine in grams that is consumed by 100 g of a chemical substance [106]: 0.1 g of carbonaceous material was added to 10 mL of 5% (*v*/*v*) HCl, boiled for 30 s, and then cooled at room temperature.

Subsequently, 100 mL of 0.1 N iodine solution (I_2_) was immediately added to the carbonaceous material and stirred for 30 s. The suspension was filtered, and 50 mL of the filtrate was titrated with 0.1 N sodium thiosulphate solution (Na_2_S_2_O_3_) using thymidine (C_12_H_22_O_11_) as an indicator [107].

The amount of iodine adsorbed per kilogram of adsorbent (IV, g/kg) was plotted against the residual iodine concentration, using logarithmic axes. If the residual iodine concentration was not within the range (i.e., 0.008–0.040 N), the procedure was repeated using different carbon masses for each isotherm point. Regression analysis was applied to the three points and the iodine number was calculated as the amount adsorbed at a residual iodine concentration of 0.02 N.

## 3. Results and Discussion

The obtained results are reported and discussed in the following section.

### 3.1. Particle Size Distribution

The particle size distribution (PSD) is a fundamental parameter to characterize the grain size of the material and provides an indication of the uniformity of the dimensions.

Figure 1 shows the grain size distribution of the three different carbon-based materials.

CARBOSORB NC 1240^®^ and AMBIOTON^®^ present a homogeneous size, where a percentage greater than 90% of the particle diameter falls between 500 and 850 µm. On the other hand, RE-CHAR^®^’s diameter ranges between 850 and 600 µm by 50%, whereas the other 50% of particles present a diameter smaller than 600 µm.

Biochar particle size is an important characteristic for its use as an amendment for soil. Lu et al. compared two different biochars with different particle sizes—fine (<0.25 mm) and coarse (<1 mm)—for their effect on soil pH and heavy metal concentration in the shoots of *S. plumbizincicola* [108]. They observed that fine biochar increased the pH of soil more than coarse. Moreover, fine biochar was more effective in reducing the concentrations of Zn in shoots than the coarse biochar, while particle size did not affect the concentrations of Cd, Cu, and Pb in the shoots of *S. plumbizincicola* [108].

Particle size results are also central in adsorption capacity. Wang et al. tested three different biochars (in their original and also reduced particle size form) and activated carbon as adsorbents of methylene adsorption [109]. Experimental results showed that dye adsorption was higher when the reduced particle size biochar was used as an adsorbent in comparison with the original particle size biochar and commercially available activated carbon. Conclusions showed that the adsorption efficiency increased with the decreasing carbon particle diameter because of the mass transfer increase [109,110]. In fact, according to Ha et al., the particle size effect appears to be more significant in the initial stage of adsorption than that of space velocity/residence time [110].

Considering the properties of the carbonaceous material analyzed in this study, RE-CHAR^®^, the finest carbonaceous material, can be a good adsorbent to improve the uptake of small molecules, such as Cu^+2^, as reported by Ali et al. [111]. This occurs because the finest carbonaceous materials generally show greater accessibility to pores. However, reducing the pore size of AMBIOTON^®^ particles can improve its performance as an adsorbent and soil improver.

### 3.2. Density, Bulk Density, and Specific Weight

Density is strongly influenced by the porosity of the material. Table 1 shows the results of the determination of the real density (ρ, kg/m^3^), the apparent density (or bulk density) (ρ_b_, kg/m^3^), and the specific weight (γ, N/m^3^).

From Table 1, it can be seen that CARBOSORB NC 1240^®^ presents the highest values for all three parameters. In fact, CARBOSORB NC 1240^®^ shows values about twice as high as AMBIOTON^®^ and four times higher than RE-CHAR^®^.

The density differences among these three materials can be explained depending on the feedstock and the pyrolysis temperature of their production process. Qambrani et al., in a review article, explained that at higher pyrolysis temperatures, volatile matter reduces, increasing the density of the solid [112]. In fact, AMBIOTON^®^ has twice the mass density value of RE-CHAR^®^ (i.e., 284 ± 3.4 and 132 ± 2.0 kg m^−3^, respectively) and has been produced at a slightly higher temperature range (i.e., 700–750 and 600–750 °C, respectively). Yang et al. also reported an increase in the particle density in rice straw (i.e., from 0.158 to 0.173 g cm^−3^) and in canola stalk biochar (i.e., from 0.155 to 0.218 g cm^−3^), when the pyrolysis temperature increases from 450 to 650 °C [113]. Moreover, raw materials of AMBIOTON^®^ and RE-CHAR^®^ are different from those of rice straw and canola stalk biochar (i.e., semi-de-husked and virgin coniferous wood, respectively), showing bulk density values slightly higher than those of rice straw and canola stalk biochar [113]. Furthermore, Pituello et al. studied biochars produced from five feedstocks (sewage sludge, municipal organic waste, cattle manure and silage digestates, poultry litter, and vineyard pruning residues) at different temperatures [114].

After physical-chemical catheterization, the results showed that for each feedstock, specific density increased according to the increase of temperature, most likely because of the conversion of low-density disordered carbon to high-density turbostratic carbon [114].

Thus, it is reasonable why CARBOSORB NC 1240^®^, which was produced at a higher temperature than AMBIOTON^®^ and RE-CHAR^®^, showed the highest density [115].

Generally, the bulk density of biochar is in the range of 470 to 600 kg/m^3^, as reported by Singh et al. [116]. Similarly, Yargicoglu et al., characterizing six different wood-derived biochars and granular activated carbon (GAC), observed that all the carbonaceous materials presented a low bulk density (<1000 kg/m^3^), reflecting the high internal porosities [117].

### 3.3. Specific Surface Area, Pore Volume, and Average Pore Size

Figure 2 and Figure 3 show adsorption and desorption isotherms of N_2_ obtained by CARBOSORB NC 1240^®^, AMBIOTON^®^, and RE-CHAR^®^.

As shown in Figure 2, both biochars presented a type IV isotherm of the IUPAC classification with an H4 hysteresis loop ranging from 0.45 to 0.98 *p*/*p*_0_, which is typical of many mesoporous industrial adsorbents. Furthermore, the profile of the hysteresis curve indicates that the pores are open tubular in shape [118]. The hysteresis loop of RE-CHAR^®^ is narrower than that of AMBIOTON^®^, so the mesopores amount of the former is inferior to that of the latter, as shown in Figure 3. Furthermore, CARBOSORB NC 1240^®^ shows a rapid increase at low relative pressure (*p*/*p*_0_ < 0.1), suggesting the existence of a good amount of micropores (0.398410 cm^3^/g), compared to AMBIOTON^®^ and RE-CHAR^®^ (0.057996 and 0.035743 cm^3^/g, respectively).

Table 2 summarizes the specific surface area (S_BET_, m^2^/g), pore volume (V_T_, cm^3^/g), and average pore size (d, Å) obtained from BET and BJH analyses, respectively.

From the adsorption of nitrogen to biochar, the following order can be drawn: CARBOSORB NC1240^®^ > RE-CHAR^®^ > AMBIOTON^®^, which shows that the activated carbon has good adsorption capacity.

The value of the specific surface of CARBOSORB NC 1240^®^ (1126.64 m^2^/g) is higher than those for AMBIOTON^®^ (256.23 m^2^/g) and RE-CHAR^®^ (280.25 m^2^/g), which was expected because of the activation process that characterized activated carbon. The specific surface area of CARBOSORB NC 1240^®^ is similar to S_BET_ (1102 m^2^/g) reported by Yu et al. for a coal-based activated carbon (F400), i.e., S_BET_ = 1030 m^2^/g, and a coconut shell activated carbon (CTIF), i.e., S_BET_ = 1156 m^2^/g [119].

Regarding biochar, Décima et al. reported a wide range of specific surface area (i.e., S_BET_ = 0.9–470.4 m^2^/g) [82]. Naghdi et al. reported a low specific surface area, i.e., S_BET_ = 47.25 m^2^/g, for pinewood biochar produced at 550 °C [120], while Chen et al. reported S_BET_ values equal to 1.82, 7.08, and 64.72 m^2^/g, depending on the pyrolysis temperature, i.e., 200, 300, and 500 °C [121], respectively.

The porosity of a material depends on the size, density, and distribution of the particles, their shape, and how they bond together [122]. The pores communicating with the external surface are named open pores and are accessible to ions or molecules [123], while the non-communicating ones are named closed pores and do not contribute to the adsorption properties of a porous material but influence its mechanical properties [124].

Both biochar and activated carbon belong to the category of mesopore media, showing a pore size between 2 and 50 nm (20–500 Å) [125,126,127]. These results are according to the values reported by Yang and Chen [128], who referred values of 5.93 and 6.23 nm.

### 3.4. Moisture Content, Volatile Matter, Ash Content, and Fixed Carbon

In Figure 4, the variation in weight (%) and derived weight (% min^−1^) of the volatile matter in CARBOSORB NC 1240^®^, AMBIOTON^®^, and RE-CHAR^®^ is shown.

The respective percentage values obtained for moisture content (M, %), volatile matter (VM, %), ash (A, %), and fixed carbon contents (FC, %) are reported in Table 3.

Four important characteristics in biochar are: (a) moisture content, (b) volatile matter, (c) ash content, which refers to the inorganic portion of biochar, and (d) fixed carbon, i.e., the portion of biochar that remains stable in the soil for a very long time or can leach into the soil or be ingested by soil microbes.

Fixed carbon is therefore generally more valuable in biochar than mobile matter [129].

As it can be seen from the results, CARBOSORB NC 1240^®^ and AMBIOTON^®^ have a similar moisture value, while that of RE-CHAR^®^ is about one-third. However, for volatile matter and ash, RE-CHAR^®^ shows the highest values, while CARBOSORB NC 1240^®^ shows the lowest. AMBIOTON^®^ appears to have intermediate values between them.

These results are in accordance with those of Yargicoglu et al., who analyzed six commercial biochars and one activated carbon. Summarizing, they highlighted a wide range of values of moisture content (i.e., M = 0.33–66.20%) and ash content (i.e., A = 1.5–65.7%) and a narrow range of values of fixed carbon (i.e., FC = 3.7–40.3%) [117]. Furthermore, the range of values of volatile substances was wider (i.e., VM = 28.0–74.1%) than that found in the materials investigated in the present study [117].

The quantities of ash in the biochar are consistent with the raw materials used in the gasification process: pine and wood derivatives are materials with a low ash content, consisting mainly of alkali metals (Ca, K, Mg), as also reported by Brewer et al. [42].

### 3.5. Conductivity

Table 4 shows the values of conductivity (G, Ω^−1^) determined for each carbonaceous material.

From the results of Table 4, it can be seen that CARBOSORB NC 1240^®^ and AMBIOTON^®^ show quite similar conductivity values, while for RE-CHAR^®^ the conductivity almost tripled as compared to AMBIOTON^®^. Therefore, among the three materials, RE-CHAR^®^ presents the highest electrical conductivity.

This parameter is representative of the salinity content and is an important characteristic in biochars as a soil improver; indeed, a high electrical conductivity is harmful to plant growth, while a low electrical conductivity can lead to the fast removal of fertilizer in the soil. Thus, electrical conductivity of biochar is crucial to select the appropriate dose of this material to add to the soil [130].

A wide range of carbonaceous materials’ conductivity was reported in the literature. For instance, Pituello et al. indicated ranges between 1.695 × 10^−6^ and 9.200 × 10^−5^ Ω^−1^ depending on the pyrolysis temperature and feedstock type [114].

### 3.6. Raman Spectroscopy

Figure 5 shows the Raman spectra of carbonaceous samples.

Looking at Figure 5, it is possible to distinguish two main peaks, D and G bands, detected at approximately 1330 and 1580 cm^−1^, respectively.

The G band is an intrinsic Raman-active band for the ideal graphite structure and corresponds to the E_2g_ vibration due to the C-C stretching in the longitudinal symmetry axis of the graphite plane. This band represents a high degree of graphitization [131].

On the other hand, band D, usually identified as A_1g_ vibration, is due to out-of-plane vibrations attributed to the presence of structural defects and assigned to disordered carbon atom-containing vacancies, impurities, or defects, such as oxygen-containing functional groups; the G band, resulting from in-plane vibrations of *sp^2^*-bonded carbon atoms, can be assigned to alkene C=C [132,133].

The ratio of intensity of D/G bands is therefore a measure of the defects present on carbon materials’ structure.

Table 5 shows the positions of the characteristic Raman bands and the integrated intensity ratios (I_D_/I_G_) of the analyzed samples.

The band intensity ratio between I_D_ and I_G_ indicates the degree of graphitization. A ratio <1 highlights a high degree of graphitization; if the ratio is >1, then a high number of functional groups are present on the surface. From Table 5, it is possible to affirm that CARBOSORB NC 1240^®^ presents the highest band intensity ratio between I_D_ and I_G_, which can be due to the activation process. Furthermore, AMBIOTON^®^ exhibits a lower I_D_/I_G_ intensity value (0.958) than RE-CHAR^®^ (1.084). This is in accordance with the fact that a biochar with a less volatile content has a more ordered carbon structure [134].

### 3.7. Scanning Electron Microscopy and Energy-Dispersive X-ray Spectroscopy

Scanning electron microscopy/energy-dispersive X-ray spectroscopy (SEM/EDS) is a microscopic technique that yields an image showing a macro-porosity physical morphology [116] and a semi-quantitative elemental composition [135].

In Figure 6, SEM images of CARBOSORB NC 1240^®^, AMBIOTON^®^, and RE-CHAR^®^ surfaces are shown, at various magnifications: 1, 10, and 30 kX.

Figure 6 shows the homogeneous and compact surface present in CARBOSORB NC 1240^®^ and the heterogeneous repetitive structure of AMBIOTON^®^ (such as cellulose structures). RE-CHAR^®^’s structure is completely different from the other two carbonaceous materials due to its small dimensions and longitudinal and transverse sections. The difference between images of biochars and CARBOSORB NC 1240^®^ is likely due to the production process.

In Figure 7, the EDS analysis graphs of selected areas on SEM of CARBOSORB NC 1240^®^ (a), AMBIOTON^®^ (b), and RE-CHAR^®^ (c) are shown.

In Table 6, the semi-quantitative elemental composition of the three carbonaceous materials is shown, through the results of energy-dispersive X-ray analysis.

Results evidence that CARBOSORB NC 1240^®^ has a much higher percentage of carbon (95.13%) than RE-CHAR^®^ (59.01%); however, RE-CHAR^®^ contains nearly twice the oxygen percentage of AMBIOTON^®^ (i.e., 3.43% and 6.69%, respectively).

Additionally, compared to the other materials, AMBIOTON^®^ contains a larger percentage of potassium (K = 8.28% for AMBIOTON^®^ vs. 1.94% and 1.45% for RE-CHAR^®^ and CARBOSORB NC 1240^®^, respectively).

Moreover, Rb was detected only in AMBIOTON^®^. This can be explained because AMBIOTON^®^ was prepared from soft and hard wood, not very resinous, which usually contain Rb as a metal, found as Rb_2_O [136].

### 3.8. X-ray Diffraction

Figure 8 shows the diffractograms of the three samples (CARBOSORB NC 1240^®^, AMBIOTON^®^, and RE-CHAR^®^, respectively).

EDS and XRD for the AMBIOTON^®^ sample showed the presence of C in high concentration (more than 80 wt.%) and small traces of Rb and K (as confirmed in the form of Rb_2_O_3_ and KO_2_ from XRD analysis). As reported in Section 3.9, the AMBIOTON^®^ was obtained by fast pyrolysis of softwood and hardwood essences (birch, willow, black locust or turkey oak, plane tree, fruit tree wood, vine, walnut), which can contain some traces of heavy elements (including Rb). Figure 9 shows XRD spectra of CARBOSORB NC 1240^®^, AMBIOTON^®^, and RE-CHAR^®^.

The large hump in the area between 20° and 30° in both biochars is due to the index of the crystalline plane C (002). The area between 42° and 47° of RE-CHAR^®^ is also very similar, which is due to the index of the crystalline plane C (002). For activated carbon CARBOSORB NC 1240^®^, the wide peak between 40° and 50° (2ϑ) is due to the axis of the graphite structure [51]. These values are in accordance with those of Huang et al. and Chen et al., who indicated that sharper peaks represent a better degree of orientation [137,138].

### 3.9. Elemental Analysis

The results of the elemental analysis are shown in Table 7 and Figure 10.

The chemical composition of carbonaceous material depends on the feedstock, pyrolysis temperature, and activation process. CARBOSORB NC 1240^®^ presents a high content of C (92.6%), which is characteristic of activated carbon prepared at high temperatures.

Indeed, Nielsen et al. reported the same C content (i.e., 92.60%) in a coconut activated carbon (S208); however, O content (i.e., 7.40%) was slightly higher compared to that found in CARBOSORB NC 1240^®^ (O = 2.67%) [139]. Moreover, N, S, P, and Na were not detected by Nielsen et al. [139]. On the other hand, Sarswat and Mohan characterized a coconut activated carbon prepared by pyrolysis at 450 °C which presented a much lower C content (i.e., 68.5%), but a higher H (i.e., 1.90%) and N content (i.e., 5.30%) [140].

AMBIOTON^®^ and RE-CHAR^®^ are produced at similar pyrolysis temperatures, between 700 and 750 °C and 600 and 750 °C, respectively, from different feedstocks, i.e., semi-dehusked wood and coniferous wood. Thus, the main composition differences are in C and O content. RE-CHAR^®^ contents were 10% higher than in AMBIOTON^®^ while O content was ten times higher in RE-CHAR^®^. RE-CHAR^®^ elemental content was similar to pinewood biochar pyrolyzed at 750 °C (i.e., C = 84.90%, H = 0.987%, and O = 9.36%). Regarding micronutrients, potassium was mostly present in AMBIOTON^®^, unlike sulfur in coal. Magnesium was present only in AMBIOTON^®^, as in the others it did not reach the limit of detection. The XRF spectrum also detected the presence of iron, calcium, and potassium only in AMBIOTON^®^.

The nitrogen content in the RE-CHAR^®^ was three times higher than in CARBOSORB NC 1240^®^, and this could be due to the different production processes [141].

Considering aromaticity (H/C) and bulk polarity (O/C), it can be assessed that CARBOSORB NC 1240^®^ presents the highest aromatic and carbonization grade because of its lowest H/C and the most polar carbon (highest O/C). These characteristics are important in the adsorption process. For instance, a study conducted by Chu et al. determined that an increase in aromatic carbon (decrease of H/C) increased the adsorption capacity of carbamazepine in water [142].

### 3.10. pH Analysis

Table 8 shows the values of pH and pH_PZC_ of the carbonaceous materials investigated.

The three carbonaceous materials are alkaline, and this can be explained by their method of production [143,144].

Zhao et al. studied the relation between the pyrolysis temperature of biochar and pH and concluded that a positive correlation between both parameters exists [145]. This can be explained because of the presence of alkaline metals such as Mg, Mn, Ca, and K in high-temperature biochars, while greater densities of acidic functional groups (phenolic and carboxylic groups) are present in materials produced at low temperatures [144,145,146].

Knowing the pH level is critical because adding alkaline biochar to an acid soil increases the earthworm population, improving soil fertility, while adding alkaline biochar to an alkaline soil damages earthworm environmental conditions (pH, EC) [144].

The isoelectric point value is representative of the external surface charges of the carbon, whereas the point of zero charges varies in response to the net total (external and internal) surface charge of the particle [147].

The pH_pzc_ is the pH at which the final value equals the initial one when the carbonaceous materials are put in contact with a solution at different pH [148]. When the pH is less than pH_zpc_, the surface of the carbonaceous material is positively charged, while at a higher pH the carbonaceous material is negatively charged [149]. Since the pH of the medium can modify the carbonaceous material and the adsorbate charge, pH_pzc_ should be considered in the adsorption process, especially for the removal of charged compounds, such as heavy metals [150].

Figure 11 shows the relationship between the pH variation and the increase in the solid–liquid ratio in the aqueous solution.

The three carbon-based sorbents presented a high pH_pzc_, and similar values were reported by To et al. in an activated carbon prepared from palm kernel shell and activated at 900 °C for 1.5 h (i.e., pH_pzc_ = 11.5) [151]. Sumalinog et al. found similar values for pyrolysis of municipal solid waste activated with KOH (i.e., pH_pzc_ = 10.0) [152].

### 3.11. Boehm Analysis

The Boehm analysis allows quantification of specific oxygen-containing surface groups on carbonaceous materials and is considered as one of the potential methods to characterize the surface functional groups of sorbent materials [153]. The results of Boehm’s titration method are shown in Table 9.

From the results presented in Table 9, it can be seen how the OH^−^ values are similar for all the materials analyzed, unlike the lactone group, which is high in CARBOSORB NC 1240^®^, low in AMBIOTON^®^, and almost absent in RE-CHAR^®^. As for the carboxyl group, it is greater in CARBOSORB NC 1240^®^ and the same in the two biochars. The higher presence of acid groups than OH^−^ groups is in perfect accordance with the high values of pH_PZC_ discussed in Section 3.10.

### 3.12. Cation and Anion Exchange Capacity

The values found for the ion, cationic (CEC), and anionic exchange capacity (AEC) analyses are shown in Table 10.

The cation exchange capacity (CEC) indicates the capacity of a material to adsorb and exchange positively charged species [154]. The CEC is a function of the presence of oxygenated functional groups in the biochar and the surface area of the material.

The results show that CARBOSORB NC 1240^®^ presented a higher CEC and AEC value than the two biochars. AMBIOTON^®^’s CEC was four times greater than that of RE-CHAR^®^, whereas the AEC of RE-CHAR^®^ was less than half that of AMBIOTON^®^.

Biochar produced at higher temperatures exhibits a high cation exchange capacity, so it is an excellent material for metal ion adsorption from aqueous media [112].

### 3.13. Methylene Blue and Iodine Index

Table 11 shows the values of the methylene blue index and iodine adsorption value.

The methylene blue index (MBI) suggests the adsorption capacity for large molecules comparable to methylene blue. MBIs of the carbonaceous materials analyzed in this study are similar and range from 25 to 35 g/kg. Comparing these values with those reported by Del Bubba et al., the CARBOSORB NC 1240^®^ MBI (30 g/kg) is similar to the MBI of a virgin mineral activated carbon (i.e., 20.0 g/kg) and a virgin vegetal activated carbon (i.e., 16.6 g/kg) [158]. On the other hand, AMBIOTON^®^ and RE-CHAR^®^ showed higher MBIs than those of sawdust biochar produced by different pyrolysis temperatures, in the range 450–850 °C (i.e., 2.2 ± 0.1 g kg^−1^ at 450 °C, 2.9 ± 0.1 g kg^−1^ at 650 °C, and 3.6 ± 0.2 g kg^−1^ at 850 °C) [158].

The iodine index is a simple and quick method to evaluate the internal surface area of the carbon and it is linked to small molecules. CARBOSORB NC 1240^®^ presented the highest IV (five times greater than AMBIOTON^®^ and RE-CHAR^®^); thus, the IVs of the carbonaceous materials analyzed in this work were similar to the specific surface area (S_BET_) (discussed in Section 3.3).

In the present study, IV values were in accordance with those of Del Bubba et al., who reported data of 102, 157, and 190 g/kg for biochars prepared at 450, 650, and 850 °C, respectively. In the same study, for a virgin mineral activated carbon and a virgin vegetal activated carbon, higher values were identified (i.e., 964 and 1192 g/kg, respectively) [158]. Moreover, Shrestha et al., for biochar prepared from rice husk and carbonized at 900 °C, reported an iodine index approximately equal to less than half the value of this study (i.e., 80 g/kg) [159].

## 4. Conclusions

The properties of carbon materials depend on two main factors: the origin of raw materials and the production process. In addition, by changing the working conditions, it is possible to obtain carbon materials with different properties, which can affect their use.

The determination of the biochar particle size is important either to decide the predisposition or not for each application or to choose the right dosage, i.e., for the use in water treatment. Since AMBIOTON^®^ and RE-CHAR^®^ showed a pore volume similar to that of commercial activated carbon, these biochars may be used not only as soil improvers but also as adsorbents. The chemical compounds present in the analyzed materials, such as oxides and carbonates (Ca, Mg, and K) present in their ash content, depend on the starting material and their partial oxidation. Among the three carbonaceous materials, CARBOSORB NC 1240^®^ may be a better adsorbent than the raw biochars (AMBIOTON^®^ and RE-CHAR^®^) due to its larger specific surface area. After activation of the biochars, the difference between the specific surface of the activated carbon and that of biochar could be reduced. Considering the characteristics of each material, it was possible to predict their performance, for example, CARBOSORB NC 1240^®^ showed better adsorption characteristics for anionic compounds, and thus it is expected that anionic pollutants will be removed preferentially by this material.

Moreover, the results of the deep characterization carried out in the present study on the selected biochars and the activated carbon show that their properties are comparable with other carbonaceous materials used either in soil or water remediation processes; therefore, they can be used for these applications. Furthermore, future experiments will be carried out to investigate the removal capability of specific contaminants by all these media.

## Figures and Tables

**Figure 1 materials-15-07162-f001:**
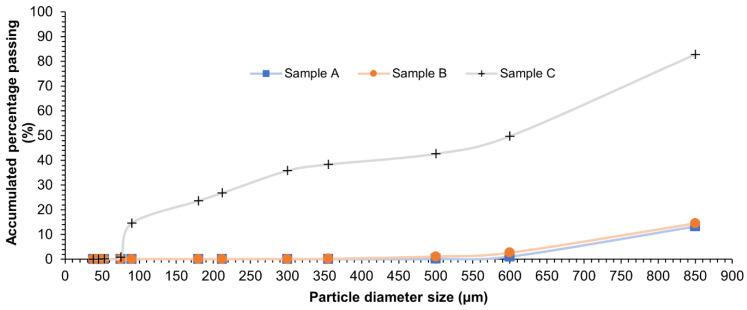
Particle size distribution (granulometric curve) of CARBOSORB NC 1240^®^ (sample A), AMBIOTON^®^ (sample B), and RE-CHAR^®^ (sample C).

**Figure 2 materials-15-07162-f002:**
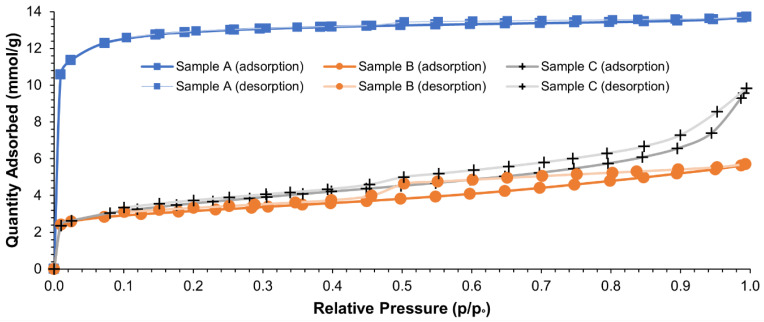
Relative pressure (*p*/*p*_0_) vs. amount of N_2_ adsorbed on CARBOSORB NC 1240^®^ (sample A), AMBIOTON^®^ (sample B), and RE-CHAR^®^ (sample C).

**Figure 3 materials-15-07162-f003:**
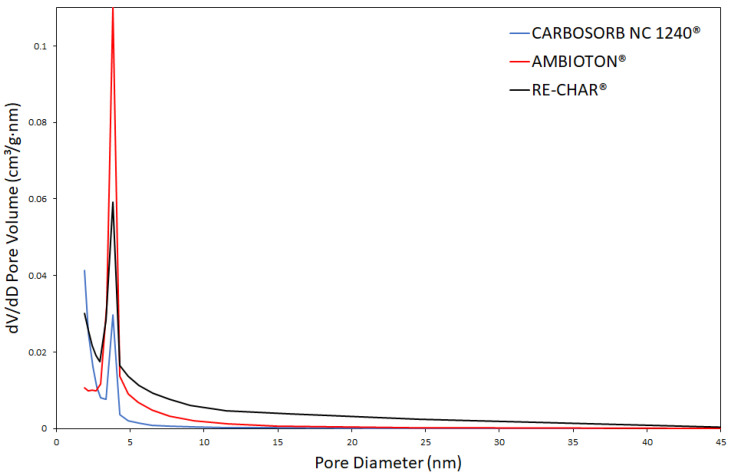
Pore size distribution of CARBOSORB NC 1240^®^, AMBIOTON^®^, and RE-CHAR^®^.

**Figure 4 materials-15-07162-f004:**
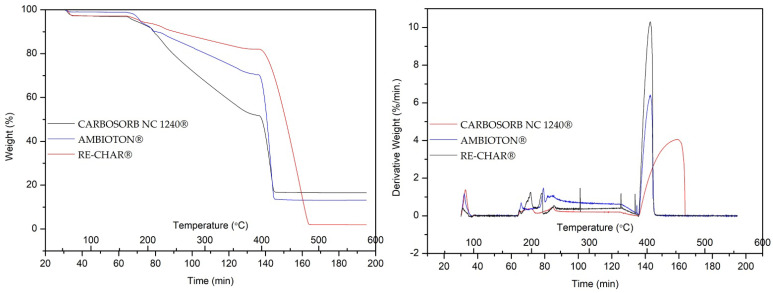
Weight (%) and derived weight (% min^−1^) of volatile matter in CARBOSORB NC 1240^®^, AMBIOTON^®^, and RE-CHAR^®^.

**Figure 5 materials-15-07162-f005:**
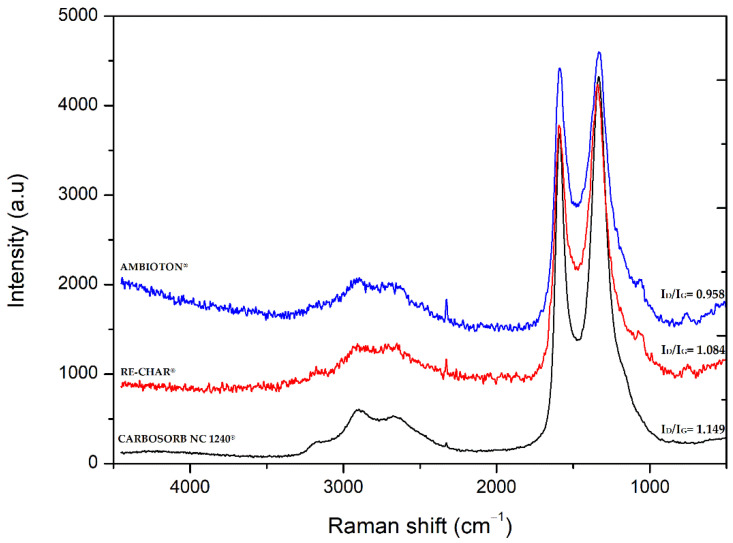
Raman spectra of carbonaceous materials. CARBOSORB NC 1240^®^, AMBIOTON^®^, and RE-CHAR^®^.

**Figure 6 materials-15-07162-f006:**
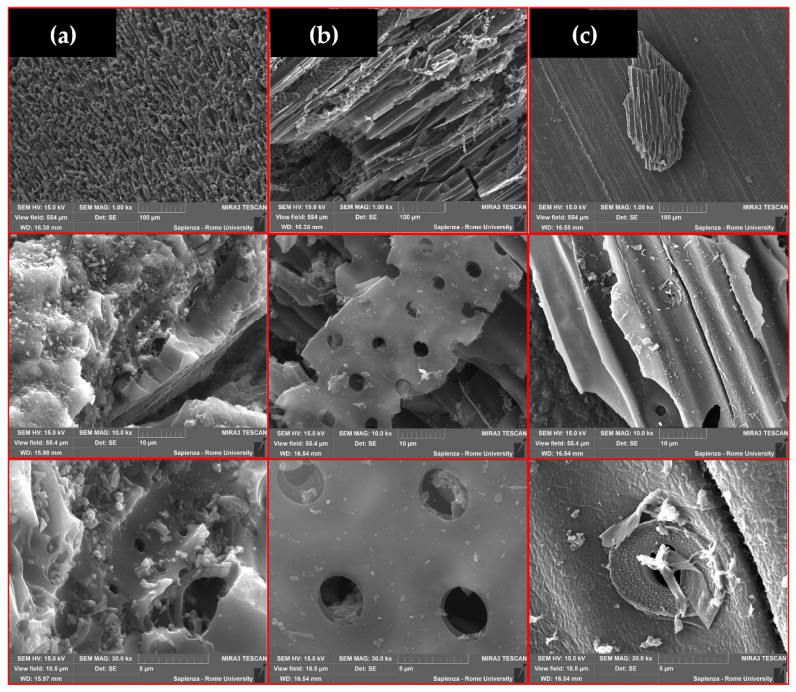
SEM images for CARBOSORB NC 1240^®^ (**a**), AMBIOTON^®^ (**b**), and RE-CHAR^®^ (**c**). Comparison with the three samples at the same magnifications (from the top: 1, 10, and 30 kX).

**Figure 7 materials-15-07162-f007:**
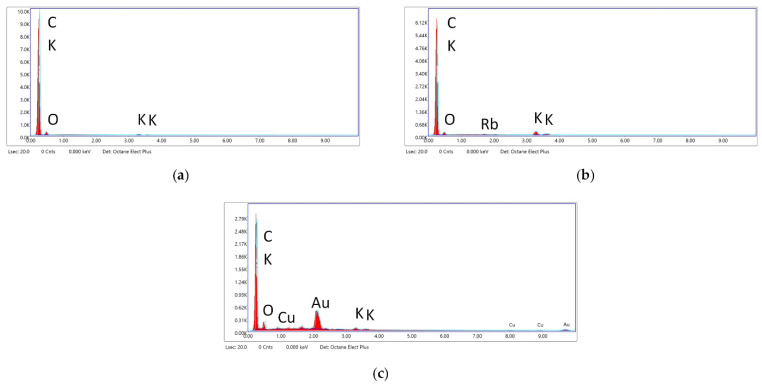
EDS analysis graph of CARBOSORB NC 1240^®^ (**a**), AMBIOTON^®^ (**b**), and RE-CHAR^®^ (**c**) (kV: 15, Mag.: 249, Take-off: 35.3, Live Time: 20 s, Amp. Time: 0.96 µs, Resolution: 130 eV).

**Figure 8 materials-15-07162-f008:**
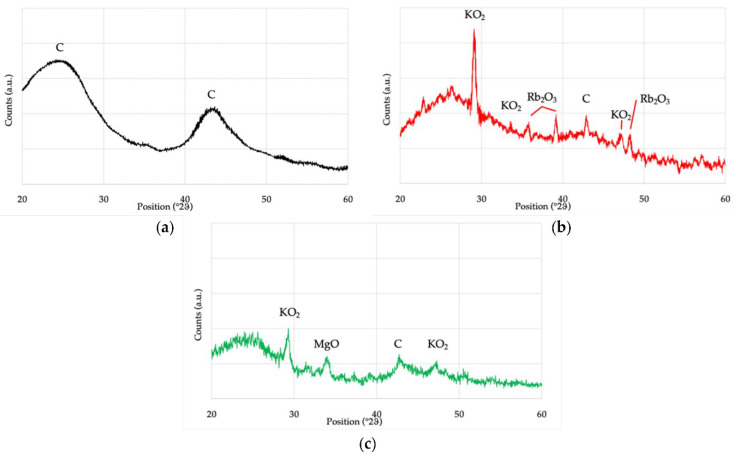
XRD spectra of CARBOSORB NC 1240^®^ (**a**), AMBIOTON^®^ (**b**), and RE-CHAR^®^ (**c**).

**Figure 9 materials-15-07162-f009:**
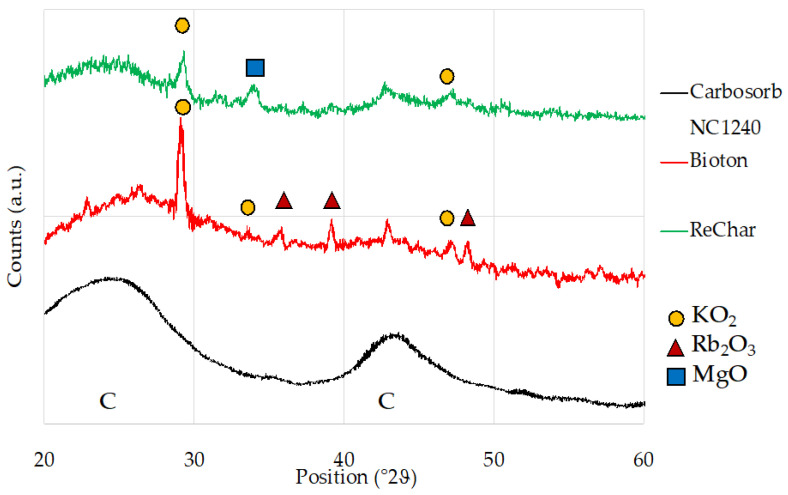
Comparison of XRD spectra of CARBOSORB NC 1240^®^, AMBIOTON^®^, and RE-CHAR^®^.

**Figure 10 materials-15-07162-f010:**
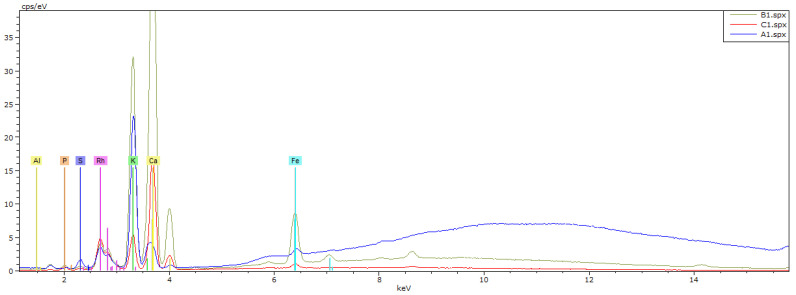
XRF analysis spectrum of CARBOSORB NC 1240^®^ (A1), AMBIOTON^®^ (B1), and RE-CHAR^®^ (C1).

**Figure 11 materials-15-07162-f011:**
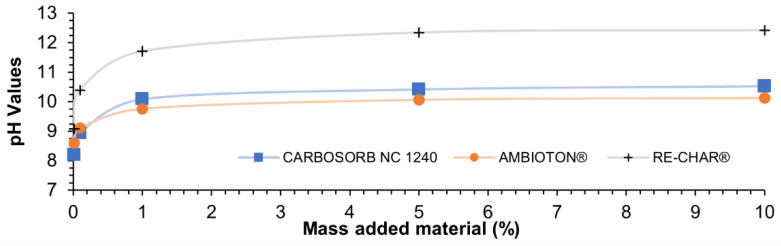
pH as a function of the added material (%).

**Table 1 materials-15-07162-t001:** Density (ρ), bulk density (ρ_b_), and specific weight (γ) of the carbonaceous materials.

	ρ (kg/m^3^)	ρ_b_ (kg/m^3^)	γ (N/m^3^)
CARBOSORB NC 1240^®^	2048.0	512 ± 4.1	20,082.69
AMBIOTON^®^	1136.0	284 ± 3.4	11,139.62
RE-CHAR^®^	528.0	132 ± 2.0	5177.57

**Table 2 materials-15-07162-t002:** BET and BJH data of carbonaceous materials.

	SSA (m^2^/g)	V_T_ (cm^3^/g)	V_micro_ (cm^3^/g)	SSA_micro_ (m^2^/g)	d (Å)
CARBOSORB NC 1240^®^	1126.64 ± 12.44	0.057 ± 0.004	0.398	1014.97	32 ± 0.60
AMBIOTON^®^	256.23 ± 9.44	0.128 ± 0.009	0.058	138.45	47 ± 1.50
RE-CHAR^®^	280.25 ± 7.12	0.28 ± 0.012	0.036	79.37	69 ± 1.10

**Table 3 materials-15-07162-t003:** Moisture content, volatile matter, ash content, and fixed carbon content of carbonaceous materials, in percentage.

	M (%)	VM (%)	A (%)	FC (%)
CARBOSORB NC 1240^®^	3.48 ± 0.20	3.22 ± 0.44	2.45 ± 0.44	90.85 ± 0.44
AMBIOTON^®^	3.18 ± 0.11	5.91 ± 0.34	6.57 ± 0.34	84.34 ± 0.34
RE-CHAR^®^	0.98 ± 0.10	6.63 ± 0.04	11.55 ± 0.44	80.84 ± 0.44

**Table 4 materials-15-07162-t004:** Conductivity (G) of the three carbon-based sorbents (Ω^−1^).

	G (S or Ω^−1^)
CARBOSORB NC 1240^®^	31.3 ± 0.4 × 10^−6^
AMBIOTON^®^	47.1 ± 0.6 × 10^−6^
RE-CHAR^®^	132.3 ± 1.7 × 10^−6^

**Table 5 materials-15-07162-t005:** Raman assignments (D and G bands) and I_D_/I_G_ ratios.

	D Band (cm^−1^)	G Band (cm^−1^)	I_D_/I_G_ ^1^
CARBOSORB NC 1240^®^	1334	1590	1.149
AMBIOTON^®^	1334	1590	0.958
RE-CHAR^®^	1341	1595	1.084

^1^ The ratio of integrated intensities is a measure of the orientation of the graphite planes and the degree of graphitization.

**Table 6 materials-15-07162-t006:** EDS analysis of the three carbon-based sorbents (elements in weight % ^1^).

	C	O	K	Rb
CARBOSORB NC 1240^®^	95.13	3.43	1.45	<LOD
AMBIOTON^®^	88.19	2.88	8.28	0.65
RE-CHAR^®^	59.00	6.69	1.94	<LOD

^1^ ZAF correction has been applied to EDAX measurements in the SEM to convert apparent concentrations (raw peak intensity) into (semi-quantitative) concentrations corrected for inter-element matrix effects. Very simplistically, Z is the atomic number correction related to the stopping power of the element, A is the absorption correction—less energetic X-rays from lighter elements are absorbed upon leaving the sample by heavier elements, and F is the fluorescence correction. A more energetic X-ray leaving the sample can fluoresce a lower-energy X-ray from a lighter element. The ZAF routine is iterative: it needs information on concentrations to proceed but these are absent at the start. Therefore, the results from the first iteration are fed back to the second and so on until a limit is reached that is statistically satisfactory.

**Table 7 materials-15-07162-t007:** Elemental composition of carbonaceous materials.

Elements (%)	CARBOSORB NC 1240^®^	AMBIOTON^®^	RE-CHAR^®^
C	92.6 ± 0.10	77.65 ± 14.1	84.5 ± 0.10
H	0.35 ± 0.02	0.9 ± 0.10	0.85 ± 0.04
N	0.05 ± 0.004	0.07 ± 0.01	0.15 ± 0.02
O	2.67 ± 0.21	1.10 ± 0.12	9.99 ± 1.05
P	0.32 ± 0.19	0.65 ± 0.19	1.79 ± 0.59
K	1.45 ± 0.13	4.57 ± 1.03	1.40 ± 0.11
S	1.88 ± 0.39	0.20 ± 0.02	0.32 ± 0.02
Ca	<LOD (0.5)	<LOD (0.5)	<LOD (0.5)
Mg	<LOD (0.5)	1.03 ± 0.03	<LOD (0.5)
O/C	0.029	0.014	0.118
H/C	0.004	0.012	0.010

**Table 8 materials-15-07162-t008:** pH and pH_PZC_ (-) of CARBOSORB NC 1240^®^, AMBIOTON^®^, and RE-CHAR^®^.

	pH (1%)	pH_PZC_
CARBOSORB NC 1240^®^	10.09 ± 0.10	10.5 ± 0.05
AMBIOTON^®^	9.76 ± 0.10	10.1 ± 0.04
RE-CHAR^®^	11.71 ± 0.10	12.4 ± 0.10

**Table 9 materials-15-07162-t009:** Boehm analysis (mmol/g) performed for the carbonaceous substances.

Materials	OH^−^	Acid	Lactone	Carboxyl	Reference
DFW Biochars	0.10	0.20	0.07	0.06	[154]
DF Biochars	0.09	0.06	0.06	0.01	
HP Biochars	0.08	0.08	0.06	0.03	
EUC-450	0.23 ± 0.011	0.63 ± 0.004	0.17 ± 0.0120	0.23 ± 0.003	[155]
EUC-600	0.09 ± 0.017	0.32 ± 0.006	0.10 ± 0.0016	0.14 ± 0.003	
Pica 150	0.64	2.51	0.75	0.87	[156]
Picaflo	0.27	1.34	0.51	0.51	
KN	0.27	0.78	0.09	0.42	[157]
CARBOSORB NC 1240^®^	0.41 ± 0.03	1.19 ± 0.09	0.450 ± 0.030	0.80 ± 0.05	This study
AMBIOTON^®^	0.40 ± 0.02	0.98 ± 0.06	0.434 ± 0.012	0.29 ± 0.03	
RE-CHAR^®^	0.44 ± 0.05	1.31 ± 0.09	<0.001	0.29 ± 0.02	

**Table 10 materials-15-07162-t010:** CEC and AEC of carbonaceous substances (cmol(+)/kg).

	CEC (cmol(+)/kg)	AEC (cmol(+)/kg)
CARBOSORB NC 1240^®^	75.0 ± 1.20	18.40 ± 0.87
AMBIOTON^®^	58.75 ± 1.40	13.02 ± 0.11
RE-CHAR^®^	12.50 ± 0.80	5.30 ± 0.45

**Table 11 materials-15-07162-t011:** Methylene blue index (MBI) and iodine index (IV) of carbonaceous materials (g/kg).

	MBI (g/kg)	IV (g/kg)
CARBOSORB NC 1240^®^	30.18 ± 1.4	1044 ± 22.1
AMBIOTON^®^	35.04 ± 2.7	211 ± 8.9
RE-CHAR^®^	25.56 ± 3.4	202 ± 12.4

## Data Availability

Not applicable.

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
