# Peer review of "Physical-Chemical Characterization of Different Carbon-Based Sorbents for Environmental Applications"

_materials, 2022, doi:10.3390/ma15207162_

Round 1

Reviewer 1 Report

Carbon-based sorbents are important for the environmental remediation processes. Biochar is a good alternative to activated carbon, providing high treatment efficiency while avoiding the industrial production of activated carbon and the accompanying environmental impacts. The research of this manuscript is interesting and results are reliable. However, major revision is required and the comments are given below.

1.     “Scanning Electron Microscope” and “Ionic Exchange Capacity” in the abstract are suggested to be revised as “scanning electron microscope” and “ionic exchange capacity” to be in accordance with “pore volume, specific surface area”.

2.     Keywords can be reduced. “adsorbent”, “natural adsorbent” and " waste adsorbent" are repeated.

3.     More references are suggested to be cited for biochar and activate carbon in the introduction part, for example Biochar 2022, 4 (1), 50.

4.     Please pay attention to the writing of units. Units should be written in the same style. For example, “mL min-1”, “°C min-1”, “m2/g” are different styles.

5.     “KV” in line 218 should be revised as “kV”. “KeV” in line 235 should be revised as “keV”.

6.     Please double check the particle size distribution curve for sample C. The percentage passing is more than 100% in total.

7.     “can be a good adsorbent to improve the uptake of small molecules, like Cu+2” in line 347-348 should be revised as “can be a good adsorbent to improve the uptake of small cations, like Cu2+”.

8.     Please pay attention to the writing of “p/p0” in line 383, Figure 2 and Figure 3.

9.     The scale should be labeled in X axial for Raman spectra in Figure 5. Please refer and cite Diamond and Related Materials 2022, 128, 109247.

10.  The horizontal and vertical scale values in some figures (for example Figure 3) are too busy.

11.  All images in the text should be carefully checked for consistency in formatting, such as Figure 10, and the lines can be bolded appropriately.

12.  Adjust the format and position of the pictures/tables in the text to keep the same format in the text and make the whole article clearer.

13.  In the conclusion part, several adsorbents should be briefly introduced, and then the results of different physical and chemical analysis should be analyzed and compared.

14.  Please pay attention to the references. Some references have doi, some lack doi.

15.  Biomass derived porous carbon materials or biochars are promising absorbents for wastewater treatment. More references are suggested to be cited, especially those newly published. Please refer and cite: New Ulva lactuca Algae Based Chitosan Bio-composites for Bioremediation of Cd(II) Ions; Synthesis and Application of Granular Activated Carbon from Biomass Waste Materials for Water Treatment: A Review.

Author Response

Carbon-based sorbents are important for the environmental remediation processes. Biochar is a good alternative to activated carbon, providing high treatment efficiency while avoiding the industrial production of activated carbon and the accompanying environmental impacts. The research of this manuscript is interesting and results are reliable. However, major revision is required and the comments are given below.

Point 1: “Scanning Electron Microscope” and “Ionic Exchange Capacity” in the abstract are suggested to be revised as “scanning electron microscope” and “ionic exchange capacity” to be in accordance with “pore volume, specific surface area”.

Response 1: Thanks for your kind suggestion. It has been corrected.

Point 2: Keywords can be reduced. “adsorbent”, “natural adsorbent” and " waste adsorbent" are repeated.

Response 2: Thanks for your kind suggestion. It has been corrected and appropriate keywords have been inserted.

Point 3: More references are suggested to be cited for biochar and activate carbon in the introduction part, for example Biochar 2022, 4 (1), 50.

Response 3: Thanks for your kind suggestion. The suggested reference has been cited in the introduction part relating to biochar and activated carbon.

Point 4: Please pay attention to the writing of units. Units should be written in the same style. For example, “mL min-1”, “°C min-1”, “m2/g” are different styles.

Response 4: Thanks for your kind suggestion. The units have been written and revised using the same style.

Point 5: “KV” in line 218 should be revised as “kV”. “KeV” in line 235 should be revised as “keV”.

Response 5: Thanks for your kind suggestion. It has been corrected.

Point 6: Please double check the particle size distribution curve for sample C. The percentage passing is more than 100% in total.

Response 6: Thanks for your kind suggestion. It has been specified that the cumulative passing percentage is shown in the graph.

Point 7: “can be a good adsorbent to improve the uptake of small molecules, like Cu+2” in line 347-348 should be revised as “can be a good adsorbent to improve the uptake of small cations, like Cu2+”.

Response 7: Thanks for your kind suggestion. It has been corrected.

Point 8: Please pay attention to the writing of “p/p0” in line 383, Figure 2 and Figure 3.

Response 8: Thanks for your kind suggestion. It has been corrected.

Point 9: The scale should be labeled in X axial for Raman spectra in Figure 5. Please refer and cite Diamond and Related Materials 2022, 128, 109247.

Response 9: Thanks for your kind suggestion. Figure 5 was labelled as suggested by the reviewer; moreover, the suggested reference has been cited.

Point 10: The horizontal and vertical scale values in some figures (for example Figure 3) are too busy.

Response 10: Thanks for your kind suggestion. Figures were labelled as suggested by the reviewer.

Point 11: All images in the text should be carefully checked for consistency in formatting, such as Figure 10, and the lines can be bolded appropriately.

Response 11: Thanks for your kind suggestion. Figures have been changed and labelled accordingly.

Point 12: Adjust the format and position of the pictures/tables in the text to keep the same format in the text and make the whole article clearer.

Response 12: Thanks for your kind suggestion.

Point 13: In the conclusion part, several adsorbents should be briefly introduced, and then the results of different physical and chemical analysis should be analyzed and compared.

Response 13: Thanks for your kind suggestion. Conclusion section was modified according to the suggestion given.

Point 14: Please pay attention to the references. Some references have doi, some lack doi.

Response 14: Thanks for your kind suggestion. References were completed with their DOI when it was possible.

Point 15: Biomass derived porous carbon materials or biochars are promising absorbents for wastewater treatment. More references are suggested to be cited, especially those newly published. Please refer and cite: New Ulva lactuca Algae Based Chitosan Bio-composites for Bioremediation of Cd(II) Ions; Synthesis and Application of Granular Activated Carbon from Biomass Waste Materials for Water Treatment: A Review.

Response 15: Thanks for your kind suggestion. The suggested references have been cited in the introduction part.

Reviewer 2 Report

The study “Physical-chemical characterization of different carbon-based sorbents for environmental applications ” explores the characteristics of materials. The study is important from the perspective of environmental applications of carbonaceous materials. It is practical; however, the following improvements are recommended.

1-      This paper carries a lot of information. Please consider organizing it so the reader can find relevant information. A figure/table might be helpful.

2-      Please justify selecting these three sorbents. Are they representative of a particular class? Are they interrelated? any other reason?

Abstract:

 Please improve the abstract. You can add some quantitative information and results. The sentence “In the latter application, it showed to represent a good alternative to activated carbon, providing high treatment efficiency with the additional advantage that its use allows to fully comply with the pillars of the circular economy since provides a solid waste (deriving from the pyrolysis process of woody materials) with a new reuse option; furthermore, industrial production of activated carbon is avoided along with the consequent environmental impacts.” is too long to understand. Also, the last two sentences are meaningful and require explanation. Please consider elaborating on them.

Introduction:

Please organize your introduction. You can follow the funnel technique. You can start broad and narrow down to your literature review, research gap, hypothesis, and approach. Also, a paragraph is three sentences or more.

Conclusions:

Please elaborate on conclusions. Consider correlating characteristics with applications i.e., which characteristics support what type of environmental applications. The readers will be interested in comparing and selecting materials for their application. 

Author Response

Point 1: The study “Physical-chemical characterization of different carbon-based sorbents for environmental applications ” explores the characteristics of materials. The study is important from the perspective of environmental applications of carbonaceous materials. It is practical; however, the following improvements are recommended.

  • This paper carries a lot of information. Please consider organizing it so the reader can find relevant information. A figure/table might be helpful.
  • Please justify selecting these three sorbents. Are they representative of a particular class? Are they interrelated? any other reason?

Response 1: Thanks for your kind suggestion. The abstract and the conclusion were completely revised according to the reviewer’s comment.

Moreover, it was added a graphical abstract and, in the introduction section (last paragraph), it was specified the importance of the three adsorbents.

Abstract

Point 2: Please improve the abstract. You can add some quantitative information and results. The sentence “In the latter application, it showed to represent a good alternative to activated carbon, providing high treatment efficiency with the additional advantage that its use allows to fully comply with the pillars of the circular economy since provides a solid waste (deriving from the pyrolysis process of woody materials) with a new reuse option; furthermore, industrial production of activated carbon is avoided along with the consequent environmental impacts.” is too long to understand. Also, the last two sentences are meaningful and require explanation. Please consider elaborating on them.

Response 2: Thanks for your kind suggestion. The abstract was completely modified to make it clearer; moreover, quantitative information was added.

Introduction

Point 3: Please organize your introduction. You can follow the funnel technique. You can start broad and narrow down to your literature review, research gap, hypothesis, and approach. Also, a paragraph is three sentences or more.

Response 3: Thanks for your kind suggestion. The introduction was completely revised according to the reviewer’s comment.

Conclusions

Point 4: Please elaborate on conclusions. Consider correlating characteristics with applications i.e., which characteristics support what type of environmental applications. The readers will be interested in comparing and selecting materials for their application.

Response 4: Thanks for your kind suggestion. Conclusion section was modified according to the suggestion given.

Reviewer 3 Report

General Comments

·         The manuscript needs a more in-depth analysis of the findings and a better discussion on the effect of physical-chemical parameters of the examined material on their adsorption capacity and their potential use as soil amendment.

·         The manuscript contains minor grammatical errors that should be corrected.

 Specific Comments

1.      Lines 344-345: Please briefly discuss the reasons behind the observation that adsorption efficiency increased with decreasing carbon particle diameter.

2.      Lines 364-365: Please explain the mechanism involved in the impact of pyrolysis temperature on the porosity and particle size of material.

3.      Figure 3:  Please comment on the significance of passage from adsorption to desorption and its impact on the adsorption capacities and characteristics of the three examined material based on the results in this figure. 

4.      Line 394: This sentence needs elaboration. How did the authors conclude from the profile of the hysteresis that the pores are open tubular in shape?

5.      Figure 4: Please identify the curves illustrating the weights and derived weights for each sample.

6.      Line 597:  Please complete the sentence (method of production?).

7.      Lines 632-633:  The comment on these lines on the quantity of the Lactone group in the sorbents do not correspond to the results presented in Table 9 which shows similar amounts of the Lactone group in AMBIOTON® and CARBOSORB NC 1240®. Please comment.

8.      Line 644: What is CSA value?

Author Response

Point 1: The manuscript needs a more in-depth analysis of the findings and a better discussion on the effect of physical-chemical parameters of the examined material on their adsorption capacity and their potential use as soil amendment.

Response 1: Thanks for your kind suggestion. The text has been fully revised.

Point 2: The manuscript contains minor grammatical errors that should be corrected.

Response 2: Thank you for the suggestion. The manuscript was fully revised and the language improved.

Point 3: Lines 344-345: Please briefly discuss the reasons behind the observation that adsorption efficiency increased with decreasing carbon particle diameter.

Response 3: Thanks for your kind suggestion. It was explained in the same paragraph

Point 4: Lines 364-365: Please explain the mechanism involved in the impact of pyrolysis temperature on the porosity and particle size of material.

Response 4: Thanks for your kind suggestion. It was explained in the same paragraph.

Point 5: Figure 3: Please comment on the significance of passage from adsorption to desorption and its impact on the adsorption capacities and characteristics of the three examined material based on the results in this figure. 

Response 5: Thanks for your kind suggestion. To better clarify the relationship between the porosity and the absorbent properties of the analyzed materials, the authors decided to replace this image with the one showing the porosimetric distribution.

Point 6: Line 394: This sentence needs elaboration. How did the authors conclude from the profile of the hysteresis that the pores are open tubular in shape?

Response 6: Thanks for your kind suggestion. The classification of adsorption hysteresis loops was recommended by the IUPAC, as reported in the following publication:

  1. Hattori, K. Kaneko, T. Ohba; Comprehensive Inorganic Chemistry II (Second Edition); 5.02 - Adsorption Properties, 2013, pag. 25-44. https://doi.org/10.1016/B978-0-08-097774-4.00502-7.

Point 7: Figure 4: Please identify the curves illustrating the weights and derived weights for each sample.

Response 7: Thanks for your kind suggestion. Figure has been changed accordingly.

Point 8: Line 597:  Please complete the sentence (method of production?).

Response 8: Thanks for your kind suggestion. It has been corrected.

Point 9: Lines 632-633: The comment on these lines on the quantity of the Lactone group in the sorbents do not correspond to the results presented in Table 9 which shows similar amounts of the Lactone group in AMBIOTON® and CARBOSORB NC 1240®. Please comment.

Response 9: Thanks for your kind suggestion. It has been corrected.

Point 10: Line 644: What is CSA value?

Response 10: Thanks for your kind suggestion. We meant the AEC value, or the anion exchange capacity; it has been corrected.

Round 2

Reviewer 1 Report

The manuscript has been well revised according to the comments and could be accepted now.

Reviewer 3 Report

The authors have responded to my questions and comments.  The manuscript can be published in the present form.